# Dietary Peppermint Extract Inhibits Chronic Heat Stress-Induced Activation of Innate Immunity and Inflammatory Response in the Spleen of Broiler Chickens

**DOI:** 10.3390/ani14081157

**Published:** 2024-04-11

**Authors:** Dandan Ma, Minhong Zhang, Jinghai Feng

**Affiliations:** State Key Laboratory of Animal Nutrition and Feeding, Institute of Animal Sciences, Chinese Academy of Agricultural Sciences, Beijing 100193, China; madandan93@163.com (D.M.); fengjinghai@caas.cn (J.F.)

**Keywords:** peppermint extract, innate immunity, inflammatory response, heat stress, DAI

## Abstract

**Simple Summary:**

Heat stress is an important stressor in poultry husbandry with the rise in global temperature. Heat stress causes immune dysfunction and inflammatory responses in broiler chickens. Supplementation with plant extracts, which can be used to improve immune function and alleviate inflammatory response, can be an effective nutritional intervention. This study evaluated the effect of peppermint extract on the morphological changes and major pattern recognition receptors (PRRs) related to innate immunity and inflammatory response in the spleen of broiler chickens under chronic heat stress. Dietary peppermint extract inhibits chronic heat stress-induced activation of innate immunity and inflammatory response in the spleen of broiler chickens. Therefore, the supplementation with peppermint extract can be used to regulate innate immunity and inflammatory response in broiler chickens.

**Abstract:**

The aim of this study was to investigate the effect of dietary peppermint extract (PE) on innate immunity and inflammatory responses in the spleen of broiler chickens under chronic heat stress. In order to further study the mechanism of the activation of innate immunity and inflammation induced by chronic heat stress and the regulatory effect of peppermint extract, we examined the spleen’s histological change, the mRNA expression of major pattern recognition receptors (PRRs) (TLR2, TLR4, NOD1, MDA5 and DAI) and transcription factors (NF-κB, AP-1 and IRF3) and downstream inflammatory cytokines (IFN-α, IFN-β, IL-1β, IL-6 and TNF-α) of innate immune signaling pathways associated with heat stress in the spleen of broiler chickens. The results indicated that chronic heat stress damaged the spleen tissue. In addition, chronic heat stress induced the activation of innate immunity and inflammatory responses by increasing the mRNA expression of TLR2, TLR4 and DAI, mRNA expression of transcriptional factors (NF-κB, AP-1 and IRF3) and the concentration of downstream inflammatory cytokines in the spleen of broiler chickens. Dietary peppermint extract alleviated the damage of spleen tissue caused by chronic heat stress. In addition, peppermint extract reduced the mRNA expression of DAI, mRNA expression of transcriptional factors NF-κB, AP-1 and IRF3, and the concentration of inflammatory cytokines in the spleen of broiler chickens under chronic heat stress. In conclusion, dietary peppermint extract could have a beneficial effect on regulating inflammatory response and innate immunity via inhibiting the activation of NF-κB, AP-1 and IRF3 signaling pathways mediated by DAI in the spleen of broiler chickens induced by chronic heat stress.

## 1. Introduction

Heat stress is one of the important environmental stressors affecting global broiler chicken production, which has harmful effects on growth performance, nutrient availability, gut microbiota composition, immune function and the welfare of broiler chickens [1,2,3]. Additionally, heat stress can cause poultry immune function dysregulation, which increases mortality and causes broiler producers to incur considerable economic losses [4,5].

Chronic heat stress alters innate immune function by inducing the overproduction of inflammatory cytokines to increase the inflammatory response. There is evidence suggesting that heat stress affects the innate immunity of the body, which could be indicated by the alteration of pro-inflammatory cytokines [6]. Previous studies have shown that heat stress decreased the concentration of IFN-γ and increased the concentration of TNF-α and IL-4 in the spleen of chickens, leading to an inflammatory response [7]. Concordantly, our own previous research also found that heat stress increased the concentration of IL-1β and IL-6, and reduced lysozyme activity in the serum of broiler chickens [8]. In combination, this research shows that chronic heat stress can activate innate immunity to increase the secretion of inflammatory cytokines and induce inflammatory response.

The innate immune response is the first barrier against viral microbial infection, which plays an important role in maintaining the health of animals such as broiler chickens [9]. Pathogen-associated molecular patterns (PAMPs) and danger-associated molecular patterns (DAMPs) are recognized by pattern recognition receptors (PRRs) in the innate immune system to induce the production of pro-inflammatory cytokines such as interleukins, interferons and tumor necrosis factors through the downstream Mitogen-activated protein kinase (MAPK) pathway, Nuclear factor-κB (NF-κB) pathway and Interferon regulatory factor 3/7 (IRF3/7) pathway [10]. Previously, we found that the Toll-like receptor signaling pathway, the NOD-like receptor signaling pathway, the RIG-I-like signaling pathway and the Cytosolic DNA-sensing signaling pathway were related to heat stress in the spleen of broiler chickens [11]. 

More than 10 Toll-like receptors (TLRs) have been found in chickens, of which TLR2 and TLR4 are the most studied. TLRs-mediated signaling pathways can be divided into MyD88-dependent and non-dependent pathways, which regulate the expression of cytokines by activating the transcription factors of NF-κB and Activator protein (AP-1) [12]. Nucleotide-binding oligomerization domain-containing protein 1 (NOD1) is the most important known pattern recognition receptor in the NOD-like receptor (NLR) family of poultry; it participates in the innate immune response through the downstream MAPK signaling pathway and the NF-κB signaling pathway. However, little research has been carried out on the role of NOD1 in poultry. Retinoic acid-inducible gene I (RIG-1)-like receptors (RLRs) include Melanoma differentiation-associated gene 5 (MDA5) [13], RIG-I (5′-Triphosphate RNA is the ligand for RIG-I) and the laboratory of genetics and physiology 2 (LGP2) [14]. MDA5 and RIG-I interact with mitochondrial virus-signaling genes (MAVS) to recruit downstream IRF-3/7 and NF-κB [15,16], resulting in the production of type I interferon and inflammatory cytokines [15,17,18]. A lack of RIG-I in chickens can result in increased sensitivity to IBV infection [19]. The DNA-dependent activator of IFN regulatory factor (DAI) receptors in the cytosolic DNA-sensing pathway can recognize DNA in cellular cytoplasm and activate the NF-κB pathway and MAPK pathway [20,21]. In addition, DAI has been shown to bind to serine/threonine TANK-binding kinase 1 (TBK1), followed by phosphorylation of IRF3 (pIRF3), which activates the transcription of interferon (IFN) type I (IFN-α and IFN-β) response genes in the nucleus [22]. The downstream transcription factors of the NF-κB and the MAPK signaling pathways are NF-κB and activator protein-1 (AP-1), respectively, which can stimulate the production of pro-inflammatory cytokines, such as IL-1β, IL-6 and TNF-α [23]. The protein family of NF-κB mainly includes p65, p50, p52, RelB and other members [24]. AP-1 is a homologous or heterodimer composed of Jun and Fos [25]. 

Peppermint is a kind of herbal plant containing many bioactive substances, including menthol, carvone, methyl acetate and pyridinone [26]. The supplementation of plant extracts may not only play an antimicrobial role but may also help to improve immune function and alleviate an inflammatory response. Many studies have shown that peppermint can alleviate the adverse effects of heat stress, thereby impacting growth performance, body temperature, gut health, immune function and carcass quality [27,28,29]. As an herbal medicine, peppermint can improve the immune response, previous research has reported that peppermint and its oil can improve immunity [30]. Research in mice has shown that menthol significantly reduces the levels of TNF-α and IL-6 in animals with gastric ulcers, which indicates menthol can protect the stomach through anti-inflammatory mechanisms [31]. Additionally, previous work has shown that menthol can suppress the inflammatory response induced by lipopolysaccharides, such as the production of inflammatory cytokines (TNF-α, IL-6 and IL-1β) and the activation of the NF-κB signaling pathway (TLR1, TNFAIP3 and NF-κB) [32]. Based on the above research findings, we hypothesized that peppermint extract could regulate the innate immunity and inflammatory response through PRRs-mediated signaling pathways in the spleen of chronically heat-stressed broiler chickens. 

## 2. Materials and Methods

### 2.1. Animal Management 

All procedures and experiments performed on broiler chickens were approved by the Animal Welfare Committee of Institutes of Animal Sciences, Chinese Academy of Agricultural Sciences (IAS, CAAS). One-day-old male Arbor Acres broiler chickens were obtained from a local hatchery and raised in an environmentally controlled room. At the age of 28 days, 144 healthy broiler chickens were randomly divided into 3 groups (Table 1) (control group, heat stress group and heat stress + peppermint extract group) with 6 replicates per group and 8 broiler chickens per replicate lasting for 14 days. Ethanol was used to extract the aerial part of mentha haplocalyx in a ratio of 10:1 and peppermint extract was obtained. The experimental dosage of peppermint extract was 300 mg/kg feed. Broiler chickens were reared in environmentally controlled chambers of the State Key Laboratory of Animal Nutrition and Feeding and were given ad libitum access to feed and water. Temperature, relative humidity and lighting programs in environmentally controlled chambers were followed by the recommendations set forth in the *Arbor Acres Broiler Handbook* from 1 to 28 days of age. During the whole experiment, the basal diet was formulated to meet the National Research Council’s (NRC, 1994) recommended requirement (Table 2). 

### 2.2. Sample Collection

At the end of the experiment, 1 bird from every replicate of 3 groups (*n* = 6/group) was euthanized by cervical dislocation, and spleens were removed and collected. One part of the spleen was fixed with 4% paraformaldehyde and stored at 4 °C for histochemical analysis. The other part of the spleen was frozen in liquid nitrogen and stored at −80 °C for further analysis. 

### 2.3. Histochemical Analysis 

Spleen tissues were fixed overnight in 4% polyformaldehyde and processed for paraffin embedding (*n* = 6/group). Then the cross-sections were cut into 6 µm sections and placed onto glass slides. The slides were dewaxed in xylene and rehydrated through gradient ethanol washed. Hematoxylin and eosin staining were performed using standard procedures. 

### 2.4. Enzyme-Linked Immunosorbent Assay (ELISA) Analysis 

For biochemical analysis, the concentrations of interleukin-1β (IL-1β, H002), interleukin-6 (IL-6, H007), tumor necrosis factor (TNF-α, H052), interferon-α (IFN-α, H023) and interferon-β (IFN-β, H024) in the spleen of broiler chickens (*n* = 6/group) were measured using chicken ELISA kits (Nanjing Jiancheng Bioengineering Institute, Nanjing, China) according to the instructions of the manufacturer. Take about 10 mg of spleen tissue, weigh it and put it into 1 ml of physiological saline. Homogenize it on a freeze grinder and centrifuge it (4 °C, 3500 rpm/min) for 20 min. Then, take the supernatant and measure cytokines according to the manufacturer’s instructions. The supernatant after centrifugation was diluted to 10% with physiological saline, and the protein content was determined using the BCA protein assay kit (Beyotime, Shanghai, China, P0010). The concentration of cytokines was determined by dividing the final measured cytokine content by 10 times the protein content. 

### 2.5. Gene Expression Analysis 

The expression of genes (NOD1, TLR2, TLR4, MDA5, DAI, p50, p65, p52, RelB, Jun, Fos and IRF3) in the spleen of broiler chickens (*n* = 6/group) was measured using real-time quantitative PCR (RT-qPCR). The primers are listed in Table 3. Trizol (Invitrogen, Waltham, MA, USA) was used to extract the total RNA from the spleen tissues of broiler chickens. The RNA concentration was detected by a Nanodrop spectrophotometer (Thermo Fisher Scientific, Waltham, MA, USA). Then, RNA was reverse transcribed into cDNA using cDNA Reverse Transcription Kits (Thermo Fisher Scientific, USA) according to the manufacturer’s instructions. RT-qPCR was performed by using an ABI 2720 Real-Time PCR machine (Applied Biosystems, Waltham, MA, USA). The amplification program was as follows: 40 cycles of 95 °C for 10 s, 60 °C for 10 s and 72 °C for 10 s. After 40 cycles, the melting curve was recorded to verify the primer specificity. The relative quantization of the target gene expression was calculated using the 2^−△△^CT method and normalized to Actin.

### 2.6. Statistical Analysis 

Differences in the expression of target genes and cytokines among the control group, heat stress group and heat stress + PE group were analyzed by one-way ANOVA followed by a least significant difference (LSD) using SPSS software 17.0. All data were presented as mean ± standard deviation. Differences with *p* < 0.05 were considered significant.

## 3. Results

### 3.1. Peppermint Extract Alleviated the Spleen Tissue Damage Induced by Chronic Heats Stress

For the control group, splenic nodules in the spleen tissue were abundant, the shape of splenic nodules was regular, the boundary between red pulps and white pulp was obvious, the histological structure was normal and no pathological changes were observed (Figure 1A).

Under the condition of heat stress, the boundary between red pulp and white pulp was blurred, no typical splenic nodule structure was found in the tissue, vascular dysplasia occurred in a large area of the spleen and vascular arrays were dense (as shown by the black arrow in Figure 1B). There were also many holes formed by focal necrosis, heterophil infiltration and loose tissue structure, as shown by the red arrow. These results indicated that heat stress caused splenic tissue injury, reduced the number of cells and resulted in inflammatory infiltration compared with the control group. 

For the heat stress + PE group, the boundary between red pulp and white pulp was clear, splenic tissue structure was dense, no necrosis was found and inflammatory infiltration was significantly alleviated (Figure 1C) compared with the heat stress group. These results indicated that peppermint extract significantly alleviated the inflammatory infiltration and tissue damage caused by heat stress.

### 3.2. Peppermint Extract Regulated the Expression of Main Pattern Recognition Receptors under Chronic Heat Stress

For the Toll-like receptor signaling pathway, heat stress significantly increased the mRNA expression level of TLR2 and TLR4 in the spleen of broiler chickens compared with the control group (*p* < 0.05). And peppermint extract had no significant effect on the mRNA expression level of TLR2 and TLR4 in the spleen of broiler chickens under chronic heat stress conditions (*p* > 0.05) (Figure 2).

For the NOD-like receptor signaling pathway, heat stress did not significantly change the mRNA expression level of NOD1 in the spleen of broiler chickens compared with the control group (*p* > 0.05) (Figure 2). And peppermint extract significantly decreased (*p* < 0.05) the mRNA expression level of NOD1 in the spleen of broiler chickens compared with the heat stress group.

For the RIG-I-like signaling pathway, heat stress did not significantly affect the mRNA expression level of MDA5 in the spleen of broiler chickens compared with the control group (*p* > 0.05). And peppermint extract had no significant effect on the mRNA expression level of MDA5 in the spleen of broiler chickens under chronic heat stress conditions (*p* > 0.05) (Figure 2).

For the Cytosolic DNA-sensing signaling pathway, heat stress significantly increased the mRNA expression level of DAI in the spleen of broiler chickens compared with the control group (*p* < 0.05). Peppermint significantly decreased the mRNA expression level of DAI in the spleen of broiler chickens under chronic heat stress conditions (*p* < 0.05) (Figure 2).

### 3.3. Peppermint Extract Reduced the Increase of Transcription Factors mRNA Expression Induced by Chronic Heat Stress 

As shown in Figure 3, heat stress significantly increased the mRNA expression level of NF-κB (p50, p65, p52 and RelB), AP-1 (Jun and Fos) and IRF3 in the spleen of broiler chickens compared to the control group (*p* < 0.05). Additionally, compared with the heat stress group, peppermint significantly decreased the mRNA expression of the transcription factors mentioned above (*p* < 0.05). 

### 3.4. Peppermint Extract Reduced the Increase of Inflammatory Cytokines Concentration Induced by Chronic Heat Stress 

Figure 4 showed that the concentrations of IL-1β, IL-6, TNF-α, IFN-α and IFN-β in the spleen of broiler chickens were significantly increased in the heat stress group compared with the control group (*p* < 0.05). Compared with the heat stress group, peppermint significantly decreased the concentration of IL-1β, IL-6, TNF-α and IFN-α (*p* < 0.05).

## 4. Discussion

A high-temperature environment can not only affect the growth performance, nutrient availability and welfare of broiler chickens but also influence the immune system [1,7]. In poultry, the major organs that are responsible for immune response are the spleen, thymus and bursa of Fabricius. The spleen, as the largest peripheral immune organ, is the second lymphoid organ and the site that connects innate immune system cells to adaptive immune system cells [33]. Previous studies have found that heat stress can damage immune organ tissues [34], affect immune organ growth and impact the overall development of broiler chickens due to decreased immune function. Concordantly, our study also found that heat stress significantly damaged spleen tissue and that experimental dosages of peppermint extract can alleviate spleen tissue damage induced by heat stress. 

Previous studies have shown that heat stress can increase the release of inflammatory cytokines such as IL-4, TNF-α, IL-1β, and IL-6 [7,8]. Research has found that heat stress significantly increased the levels of serum IL-1β, IL-6 and IL-18 in broilers [35]. In the study of hens, it was found that after 7 and 14 days of heat stress treatment, the mRNA expression of inflammatory cytokines (IL-6 and IL-18) in the hypothalamus significantly increased [36]. It was found in yellow-feather broilers that heat stress significantly decreased the index (organ weight/body weight) of the spleen, thymus and bursa Fabricius, and significantly increased the mRNA expression levels of IL-1β, IL-4, IL-6 and TNF-α of the spleen [35]. Our study also found that heat stress can increase the concentration of IL-1β, IL-6, and TNF-α in the spleen of broiler chickens. Previous study investigated the effect of peppermint powder on the immune system of broiler chickens under heat stress and found that dietary supplementation of 2% peppermint powder increased IgG and IgM serum levels and significantly increased the number of white bold cells, thereby enhancing immune function [37]. In the present study, we found that peppermint extract significantly decreased the release of inflammatory cytokines under chronic heat stress conditions. Although these studies, in combination, show that peppermint has both antimicrobial and immune enhancement effects, relatively little work has been done on the molecular mechanism by which peppermint extract can impact innate immunity and inflammatory response. These results indicated that peppermint extract reduced the release of inflammatory cytokines, alleviated spleen tissue injury and mitigated the inflammatory response under heat stress. 

Research has shown that the level of TLR4 mRNA increases significantly in the peripheral blood mononuclear cells of heat-stressed pigs, which shows that heat stress might regulate host immune response by regulating the expression of TLR4 [38]. Likewise, heat stress was shown to increase the expression of TLR4, IL-6 and IL-8 in the ileum and jejunum but decreased the level of IL-8 in the plasma of broiler chickens [39]. Another study found that heat stress upregulated TLR2, TLR4 and IL-6 production, and activated the NF-κB, p38 kinase and ERK signaling pathways in human monocytes [40]. Also, NOD receptors have been shown to be involved with heat stress in carp, rainbow trout, cattle with tumors, beef cattle and lactating cattle, and heat stress significantly increased the expression of TLR2 and TLR4 in gill, liver, kidney and blood tissues [41]. Consistent with these studies, our research also found that the mRNA expression levels of TLR2 and TLR4 in the spleen of broiler chickens were significantly upregulated under the condition of heat stress. Peppermint, as an anti-inflammatory agent, can inhibit the production of pro-inflammatory mediators by downregulating the NF-κB and MAPK pathways, thereby suppressing inflammation [42]. Moreover, previous studies reported that menthol could suppress the inflammatory response induced by lipopolysaccharides, such as the concentration of inflammatory cytokines (IL-1β, TNF-α and IL-6) and the activation of the NF-κB signaling pathway (TLR1, TNFAIP3 and NF-κB) [32]. So far, there is no literature on the effect of peppermint extract on the IRF signaling pathway. Our research findings not only found that the peppermint extract downregulated NF-κB and AP-1 signaling pathways but also downregulated the IRF3 signaling pathway in the spleen of broiler chickens under chronic heat stress. Furthermore, the downregulation of NF-κB, AP-1 and IRF3 signaling pathways by peppermint extract under chronic heat stress is likely mediated by pattern recognition receptors. 

The DNA-dependent activator of IFN regulatory factors (DAI), also known as DLM-1/ZBP1, functions as a DNA sensor, which can recognize DNA in the cytoplasm to activate the innate immune response mediated by the NF-Κb pathway, MAPK pathway and IRF3 pathway [20,21,22,43]. The effect of heat stress on the DAI and the regulatory effect of peppermint extract, to our knowledge, has not yet been investigated. In the present study, we found that heat stress significantly upregulated the mRNA expression of DAI in the spleen of broiler chickens. In addition, heat stress can lead to impaired permeability, making it susceptible to infection and inflammation, as well as altering the composition and abundance of the microbiome [44]. Furthermore, it may affect the expression of pattern recognition receptors through the gut–spleen axis [45]. We found, for the first time, that dietary peppermint extract significantly decreased the mRNA expression of DAI, and decreased the mRNA expression of transcription factors (NF-κB, AP-1 and IRF3), thereby decreasing the concentration of inflammatory cytokines to inhibit activation of innate immunity and inflammatory response induced by chronic heat stress in the spleens of broiler chickens. These findings also validated our hypothesis that peppermint extract could regulate the innate immunity and inflammatory response through PRRs-mediated signaling pathways in the spleen of chronically heat-stressed broiler chickens (Figure 5).

## 5. Conclusions

In summary, we found that chronic heat stress activated the TLR2, TLR4 and DAI-mediated signaling pathways, thereby activating innate immunity and inflammatory response in the spleen of broiler chickens. Meanwhile, their downstream NF-κB, AP-1 and IRF3 signaling pathways were also activated in these processes and elevated the concentration of pro-inflammatory cytokines. In addition, dietary supplementation with peppermint extract could inhibit NF-κB, AP-1 and IRF3 signaling pathways mediated by DAI in the spleen of broiler chickens under chronic heat stress conditions (Figure 5). Thus, peppermint extract could alleviate chronic heat stress-induced activation of innate immunity and inflammatory response in broiler chickens. These findings provide a theoretical basis for the use of a natural agent, peppermint extract, in the regulation of innate immunity and inflammatory response of broiler chickens under chronic heat stress conditions. 

## Figures and Tables

**Figure 1 animals-14-01157-f001:**
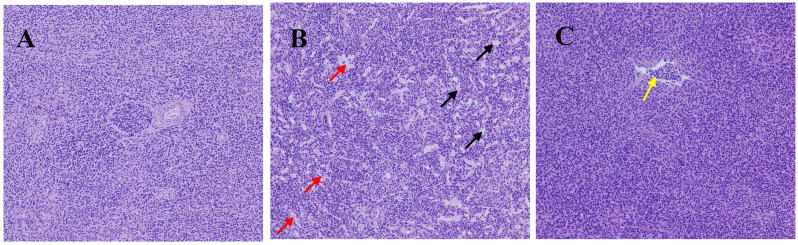
Histological examination of the spleen in broiler chickens of the control group, heat stress group and heat stress + PE group (*n* = 6/group). Representative images were captured by a microscope (NIKON DS-U3) at 200× (**A**) control group; (**B**) heat stress group; (**C**) heat stress + PE group). Black arrow: the boundary between the red and white pulp within the tissue is blurred, and no typical splenic nodule structure is observed; large areas of abnormal proliferation of blood vessels in the spleen are densely arranged and significantly increased in number. Red arrows: multiple small focal necrotic cavities can be seen in the spleen, accompanied by a small amount of heterophil infiltration and loose tissue structure. Yellow arrow: the white pulp within the tissue diffuses throughout the entire spleen, and the boundary between the red and white pulp is unclear; in the red pulp, some inflammatory cells in the blood vessels can be seen to aggregate into clusters (mainly lymphocytes).

**Figure 2 animals-14-01157-f002:**
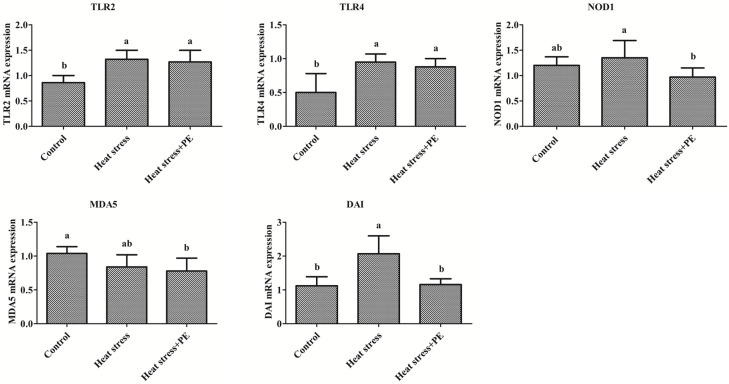
Main pattern recognition receptors mRNA expression levels in the spleen of broiler chickens of the control group, heat stress group and heat stress + PEt group (*n* = 6/group). All data were presented as mean ± standard deviation. There were significant differences between different lower-letters (a and b) (*p* < 0.05), but no significant differences between the same lower-letters (*p* < 0.05).

**Figure 3 animals-14-01157-f003:**
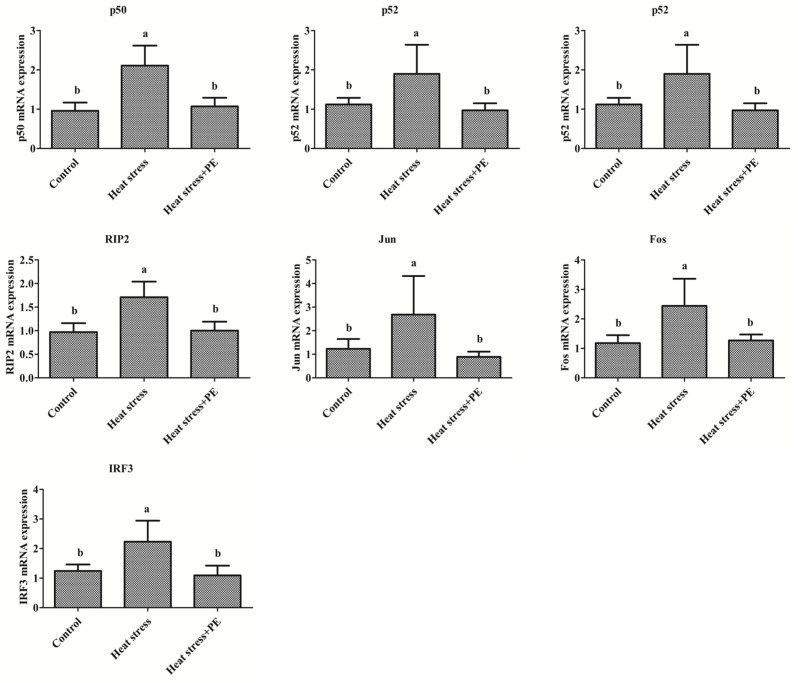
Transcription factors mRNA expression levels in the spleen of broiler chickens of the control group, heat stress group and heat stress + PE group (*n* = 6/group). All data were presented as mean ± standard deviation. There were significant differences between different lower-letters (a and b) (*p* < 0.05), but no significant differences between the same lower-letters (*p* > 0.05).

**Figure 4 animals-14-01157-f004:**
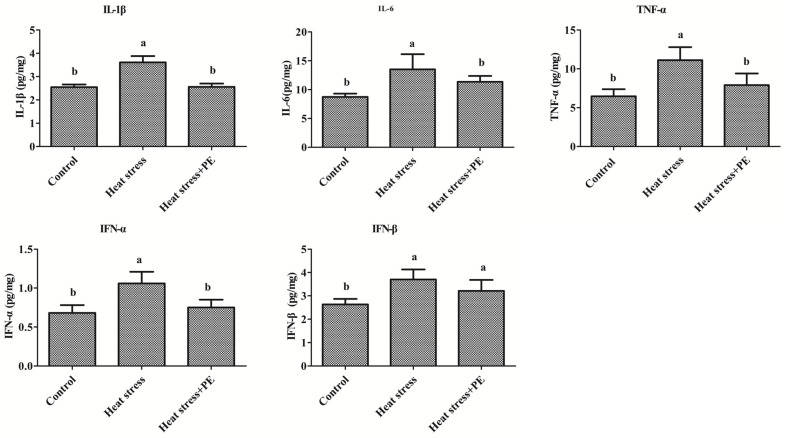
Inflammatory cytokines concentration in the spleen of broiler chickens of the control group, heat stress group and heat stress + PE group (*n* = 6/group). All data were presented as mean ± standard deviation. There were significant differences between different lower letters (a and b) (*p* < 0.05), but no significant differences between the same lower letters (*p* > 0.05).

**Figure 5 animals-14-01157-f005:**
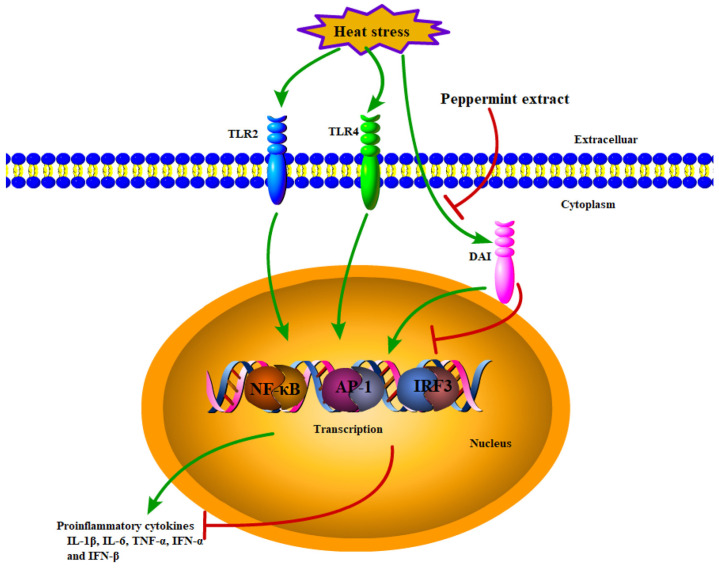
Schematic diagram of the possible molecular mechanism of peppermint extract’s regulation of innate immunity and inflammatory response in the spleen of broiler chickens under heat stress conditions. Chronic heat stress promoted the release of inflammatory cytokines by activating pattern recognition receptors TLR2, TLR4 and DAI, and their downstream signaling pathways of NF-κB, AP-1 and IRF3, which ultimately led to the activation of innate immunity and inflammatory response (green arrows). Moreover, peppermint extract reduced the release of inflammatory cytokines by down-regulating the expression of NF-κB and AP-1 mediated by DAI to inhibit the activation of innate immune and inflammatory responses in the spleen of broiler chickens caused by heat stress (red arrows).

**Table 1 animals-14-01157-t001:** Grouping of the experiment.

Group	Temperature (°C)	Diet
Control group	21 °C	Basal diet
Heat stress group	31 °C	Basal diet
Heat stress + PE group	31 °C	Basal diet + peppermint extract

**Table 2 animals-14-01157-t002:** Ingredients and nutritional components of the basal diet.

Ingredients (g/kg)	Content (%)	Ingredients (g/kg)	Content (%)
Corn	56.51	Calculated nutrient levels	
Soybean meal	35.52	Metabolizable energy (MJ/kg)	12.73
Soybean oil	4.50	Crude protein (g/kg)	20.07
NaCl	0.30	Available phosphorus (g/kg)	0.40
Limestone	1.00	Calcium (g/kg)	0.90
Dicalcium phosphate	1.78	Lysine (g/kg)	1.00
DL-Methionine	0.11	Methionine (g/kg)	0.42
Premix ^1^	0.28	Methionine + cysteine (g/kg)	0.78
Total	100.00		

^1^ Premix provided the following per kg of the diet: vitamin A, 10,000 IU; vitamin D3, 3400 IU; vitamin E, 16 IU; vitamin K3, 2.0 mg; vitamin B1, 2.0 mg; vitamin B2, 6.4 mg; vitamin B6, 2.0 mg; vitamin B12, 0.012 mg; pantothenic acid calcium, 10 mg; nicotinic acid, 26 mg; folic acid, 1 mg; biotin, 0.1 mg; choline, 500 mg; Zn (ZnSO_4_·7H_2_O), 40 mg; Fe (FeSO_4_·7H_2_O), 80 mg; Cu (CuSO_4_·5H_2_O), 8 mg; Mn (MnSO_4_·H_2_O), 80 mg; I (KI) 0.35 mg; Se (Na_2_SeO_3_), 0.15 mg.

**Table 3 animals-14-01157-t003:** Primer sequence for real-time quantitative PCR.

Gene Name	Accession Number	Primer Sequences (5′-3′)	Base Number	Product Length (bp)
β-actin	NM_205518.1	Forward: TCCACCGCAAATGCTTCTAA Reverse: GGGGCGTTCGCTCCA	20 15	205
NOD1	NM_001318438.1	Forward: AGGAGGTCTCATCAGCGAACATCT Reverse: GCAGCCTCAGCAGAAGAGCATT	24 22	217
TLR2	NM_204278.1	Forward: CCTGGTGTTCCTGTTCATCCTCAT Reverse: AGTTGGAGTCGTTCTCACTGTAGG	24 24	173
TLR4	NM_001030693.1	Forward: ACGGAAGGCTTTGGTTGGGATT Reverse: GATGTTGCTATCTGGTGCTTGGAA	22 24	184
MDA5	NM_001193638.1	Forward: GTGGCTTCAAGTGGCTCAGGAG Reverse: TCTTCTGGCGGCATCTCTTGGA	22 22	107
DAI	NM_205071.1	Forward: CAAGCGGTTGGTGCCATCATTG Reverse: ATCCTGCCTTGTGCCTTGAACTG	22 23	167
RIP2	NM_001030943.1	Forward: AGCCGCACCTGAGGAACAAGA Reverse: CCGTTGCTGGACTGGATGATGAG	22 24	201
Jun	NM_001031289.1	Forward: AGCCGCACCTGAGGAACAAGA Reverse: CCGTTGCTGGACTGGATGATGAG	21 23	109
Fos	NM_205508.1	Forward: GACTTAGCAACGACCCATCTTACG Reverse: CAGAACATTCAGACCACCTCAACA	24 24	155
p50	NM_001347945.1	Forward: GCCGACACGCAGAGCAAGAT Reverse: ACAACAGCCAGGTTCTCCTTCATT	20 24	248
p65	NM_205129.1	Forward: TCATCCACCGCCGCCACATT Reverse: GGCTGAGGAAGGCACTGAAGTC	20 22	232
p52	NM_001159511.1	Forward: TCTACCGTGAACTGGAACAGAACA Reverse: CTGGACACTAAGACTGCTGCTATG	24 24	251
RelB	D13794.1	Forward: CGGCACAGCTTCAGCAACCT Reverse: TCACCACGTTCATATCCACCTCCT	20 24	145
IRF3	NM_205372.1	Forward: ACCGCCGTATCTTCCGCATC Reverse: GGTCCTCCAGCAGCATGAACAT	20 22	192

## Data Availability

Data are contained within the article.

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
