# Peer review of "Dietary Peppermint Extract Inhibits Chronic Heat Stress-Induced Activation of Innate Immunity and Inflammatory Response in the Spleen of Broiler Chickens"

_animals, 2024, doi:10.3390/ani14081157_

Round 1
Reviewer 1 Report
Comments and Suggestions for Authors
This study examined the effect of peppermint extract on the heat stress-activated inflammatory responses in broiler chickens. The study is well-designed, and the discussion is well-written based on the obtained data. However, the manuscript needs improvements, especially in the methodology sections, with clarification and further explanation as follows:
1. The background of the study should be described in the abstract.
2. The introduction needs to be revised. Repetitive explanations and unclear descriptions can be seen in this section. The authors should describe the detailed downstream pathways, including which transcription factor leads to which cytokine expressions, whether DAI induces AP-1 and IRFs, and the function of IRFs.
3. The materials and methods section lacks the detailed information. How did the authors prepare the spleen samples for ELISA? Did the authors use the same weights of spleens to normalize? or did the authors measure the total protein level adjustment? How did the authors synthesize cDNA from extracted RNA? The authors should add appropriate references or gene accession numbers in Table 2. The statistical analysis needs improvement. Did the authors perform tests for normality/ equality of variances? Which post hoc test did the authors use when the data showed significant differences in the ANOVA test.
4. Line 38 and 47: The description of “body’s” should be removed.
5. Line 41-42: The authors should mention in which tissue/serum those results were obtained.
6. Line 68: Appropriate references should be added.
7. Line 110: Total numbers of samples used for the analysis/assay should be specified in the materials and methods and in the figure legends.
8. Line 152: Chickens do not have neutrophils.
9. Line 178-179: MDA “5”
10. Line 214: Spleen is the second lymphoid organ and the site to connect innate immune system cells to adaptive immune system cells (Mebius and Kraal, 2005). I disagree with the description of “the main site of innate immune response”.
11. Line 230 “Although these studies”: Please indicate which studies.
12. Line 258-260: The authors should revise this sentence as they used the exactly same sentence in the introduction (line 68-69).
13. Line 265: Please enumerate the name of the transcription factors.
14. Figure 1: Please explain which color of arrow shows which structure.
15. Figure 2-4: The number of samples (n = xx) and how the data are presented (mean + standard deviation? Standard error?) should be described in the figure legends. The letters in the graphs are unreadable. Larger font size should be used.
16. Figure 5: This figure misleads readers as the red arrows look like showing that DAI inhibits transcription factors and transcription factors inhibit cytokine expressions. Please revise it.
Comments on the Quality of English Language
The manuscript has some grammatical errors.
Please spell out abbreviations when the authors use the abbreviation at the first time.
Author Response
Response to reviewer 1
Dear reviewer:
Thank you very much for taking the time to review my manuscript. Please find the detailed responses below and the corresponding revisions highlighted in yellow of the re-submitted files.
Point-by-point response to comments and suggestions for authors
Comment 1: The background of the study should be described in the abstract.
Response 1: The editor asked me to supplement a simple summary before the abstract. Thus, I supplemented the study background in the simple summary.
Comment 2: The introduction needs to be revised. Repetitive explanations and unclear descriptions can be seen in this section. The authors should describe the detailed downstream pathways, including which transcription factor leads to which cytokine expressions, whether DAI induces AP-1 and IRFs, and the function of IRFs.
Response 2: I have added the description of downstream pathways, including “In addition, DAI has been shown to bind to serine/threonine TBK1, followed by phosphorylation of IRF3 (pIRF3), which activates the transcription of interferon (IFN) type Ⅰ (IFN-α and IFN-β) response genes in the nucleus [48]. The downstream transcription factors of the NF-κB and the MAPK signaling pathways are NF-κB and activator protein-1 (AP-1), respectively, which can stimulate the production of pro-inflammatory cytokines, such as IL-1β, IL-6 and TNF-α [49].”.
Comment 3: The materials and methods section lacks the detailed information. How did the authors prepare the spleen samples for ELISA? Did the authors use the same weights of spleens to normalize? or did the authors measure the total protein level adjustment? How did the authors synthesize cDNA from extracted RNA? The authors should add appropriate references or gene accession numbers in Table 2. The statistical analysis needs improvement. Did the authors perform tests for normality/ equality of variances? Which post hoc test did the authors use when the data showed significant differences in the ANOVA test.
Response 3: I have supplemented the Materials and Methods.
In the section of Enzyme-Linked Immunosorbent Assay (ELISA) Analysis, we used the same weight of spleen and measured the total protein level, and supplemented the information of ELISA kits.
I supplemented the synthesized cDNA from extracted RNA in the section of Gene Expression Analysis and the accession number for target genes in the Table 2.
In the section of Statistical Analysis, we supplemented “Differences of expression of target genes and cytokines among control group, heat stress groups and heat stress+PE group were analyzed by One-way ANOVA followed by a least significant difference (LSD) using SPSS software 17.0. All data were presented as mean ± standard deviation. Differences with P <0.05 were considered significant”.
Comment 4: Line 38 and 47: The description of “body’s” should be removed.
Response 4: The description of “body’s” has been removed. Modification as follows: Chronic heat stress changes innate immune function by inducing the overproduction of inflammatory cytokines to increase the inflammatory response. The innate immune response is the first barrier against viral microbial infection, which plays an important role in maintaining the health of animals such as broiler chickens [15].
Comment 5: Line 41-42: The authors should mention in which tissue/serum those results were obtained.
Response 5: I have added the tissue. Modification as follows: Previous study has shown that heat stress decreases the concentration of IFN-γ and increases the concentration of TNF-α and IL-4 in spleen of chickens, leading to inflammatory response.
Comment 6: Line 68: Appropriate references should be added.
Response 6: I have added the reference. Modification as follows: A lack of RIG-I in chickens can result in increased sensitivity to IBV infection [39].
Comment 7: Line 110: Total numbers of samples used for the analysis/assay should be specified in the materials and methods and in the figure legends.
Response 7: I have specified the total numbers of samples in the materials and methods and in the figure legends.
Comment 8: Line 152: Chickens do not have neutrophils.
Response 8: Through literature review, it was found that chickens have neutrophils. The articles are as follows: Yang Z, Wang S, Yin K, Zhang Q, Li S. MiR-1696/GPx3 axis is involved in oxidative stress mediated neutrophil extracellular traps inhibition in chicken neutrophils. J Cell Physiol. 2021 May;236(5):3688-3699. doi: 10.1002/jcp.30105. Epub 2020 Oct 12. PMID: 33044016; Zhu H, Yu Q, Ouyang H, Zhang R, Li J, Xian R, Wang K, Li X, Cao C. Antagonistic Effect of Selenium on Fumonisin B1 Promotes Neutrophil Extracellular Traps Formation in Chicken Neutrophils. J Agric Food Chem. 2022 May 18;70(19):5911-5920. doi: 10.1021/acs.jafc.2c01329. Epub 2022 May 10. PMID: 35535747.
Comment 9: Line 178-179: MDA “5”
Response 9: I have made the modification. Modification as follows: For the RIG-I-like signaling pathway, heat stress did not significantly affect the mRNA expression level of MDA5 compared with control group (P >0.05). And peppermint extract had no significant effect on the mRNA expression level of MDA5 under chronical heat stress condition (P >0.05) (Figure 2).
Comment 10: Line 214: Spleen is the second lymphoid organ and the site to connect innate immune system cells to adaptive immune system cells (Mebius and Kraal, 2005). I disagree with the description of “the main site of innate immune response”.
Response 10: I have made modifications to this sentence and added new reference. Modification as follows: The spleen is an important lymphoid organ and the site to connect innate immune system cells to adaptive immune system cells [40].
Comment 11: Line 230 “Although these studies”: Please indicate which studies.
Response 11: The “these studies” refers to the studies mentioned earlier.
Comment 12: Line 258-260: The authors should revise this sentence as they used the exactly same sentence in the introduction (line 68-69).
Response 12: I have deleted this sentence due to duplication.
Comment 13: Line 265: Please enumerate the name of the transcription factors.
Response 13: I have enumerated the name of the transcription factors. Modification as follows: We also found, for the first time, that dietary peppermint extract significantly decreased the mRNA expression of DAI, and decreases the mRNA expression of transcription factors (NF-κB, AP-1 and IRF3), thereby decreasing the concentration of inflammatory cytokines to inhibits activation of innate immunity and inflammatory response induced by chronical heat stress in spleens of broiler chickens.
Comment 14: Figure 1: Please explain which color of arrow shows which structure.
Response 14: I have added the explanations of black, red and yellow arrow. Modification as follows: Black arrow: the boundary between the red and white pulp within the tissue is blurred, and no typical splenic nodule structure is observed; large areas of abnormal proliferation of blood vessels in the spleen are densely arranged and significantly increased in number. Red arrows: multiple small focal necrotic cavities can be seen in the spleen, accompanied by a small amount of neutrophil infiltration and loose tissue structure. Yellow arrow: the white pulp within the tissue diffuses throughout the entire spleen, and the boundary between the red and white pulp is unclear; in the red pulp, some inflammatory cells in the blood vessels can be seen to aggregate into clusters (mainly lymphocytes).
Comment 15: Figure 2-4: The number of samples (n = xx) and how the data are presented (mean + standard deviation? Standard error?) should be described in the figure legends. The letters in the graphs are unreadable. Larger font size should be used.
Response 15: I have added the number of sample and “All data were presented as mean ± standard deviation” in the figure legends. In addition, I enlarged the letters in the figures.
Comment 16: Figure 5: This figure misleads readers as the red arrows look like showing that DAI inhibits transcription factors and transcription factors inhibit cytokine expressions. Please revise it.
Response 16: The explanation of the red arrow has been placed in the figure legend. The red arrows indicated that peppermint extract reduced the release of inflammatory cytokines by down-regulating the expression of NF-κB and AP-1 mediated by DAI to inhibit the activation of innate immune and inflammatory response in spleen of broiler chickens caused by heat stress (red arrows).
Response to comments on the quality of English language
Comment: The manuscript has some grammatical errors.
Please spell out abbreviations when the authors use the abbreviation at the first time.
Response: I have spelt out abbreviations when I used the abbreviation at the first time.
Reviewer 2 Report
Comments and Suggestions for Authors
This study investigated the effects of peppermint extract (PE) in feed on innate immunity and inflammation in the spleen of broiler chickens under chronic heat stress. Histological changes, mRNA expression of pattern recognition receptors and transcription factors, and levels of inflammatory cytokines in the spleen of heat-stressed broiler chickens revealed that chronic heat stress damaged splenic tissues and activated innate immune and inflammatory responses. Peppermint extract attenuated splenic tissue damage and reduced mRNA expression of key factors and inflammatory cytokine levels. Overall, this manuscript is well thought out, well presented, and original, but there are some minor issues.
Question 1.
Line 2: The content of the manuscript does not fully support the title and suggests an adjustment to the title
Question 2.
line90: Lack of tables of components and nutrient levels of broiler diets
Question 3.
Line 101: Why is PE added at 300mg/kg, is there a reference?
Question 4.
The experimental design took three control groups, including Control group, Heat stress group and Heat stress+PE group, and it is recommended to add Control group+PE group (with different concentrations) so that it can reflect the suitability of economical broilers to PE.
Author Response
Response to reviewer 2
Dear reviewer:
Thank you very much for taking the time to review my manuscript. Please find the detailed responses below and the corresponding revisions highlighted in yellow of the re-submitted files.
Point-by-point response to comments and suggestions for authors
Comment 1: Line 2: The content of the manuscript does not fully support the title and suggests an adjustment to the title
Response 1: I changed the title to “Dietary Peppermint Extract Inhibits Chronical Heat Stress-Induced Activation of Innate Immunity and Inflammatory Response In Spleen of Broiler Chickens”.
Comment 2: line90: Lack of tables of components and nutrient levels of broiler diets
Response 2: I supplemented the ingredients and nutrients of the basal diet in Table 2.
Comment 3: Line 101: Why is PE added at 300mg/kg, is there a reference?
Response 3: The 300mg/kg was obtained from our laboratory’ s previous heat stress experiment to alleviate heat stress in broiler chickens, but it has not been published in SCI and is only included in master’s thesis (常双双.偏热环境对肉鸡盲肠菌群多样性、挥发性脂肪酸和血清脑肠肽影响及薄荷提取物对其调控作用研究[D].河北工程大学,2018.).
Comment 4: The experimental design took three control groups, including Control group, Heat stress group and Heat stress+PE group, and it is recommended to add Control group+PE group (with different concentrations) so that it can reflect the suitability of economical broilers to PE.
Response 4: The 300mg/kg was obtained from our laboratory’ s previous heat stress experiment to alleviate heat stress in broiler chickens, but it has not been published in SCI and is only included in master’s thesis (常双双.偏热环境对肉鸡盲肠菌群多样性、挥发性脂肪酸和血清脑肠肽影响及薄荷提取物对其调控作用研究[D].河北工程大学,2018.). Therefore, we did not set up groups with different concentrations of PE and directly used the most suitable addition amount obtained in the laboratory before.
Reviewer 3 Report
Comments and Suggestions for Authors
Dear author: your manuscript "Dietary Peppermint Extract Inhibits Chronical Heat Stress-Induced Activation of Innate Immunity and Inflammatory Response Via DAI in Spleen of Broiler Chickens" aligns well with the journal guidelines, demonstrating a thorough understanding of the recommended structure and content. However, there are some minor and major suggestions for improvement across various sections:
Minor suggestions:
The well-written introduction provides a clear overview of the study's context. Consider briefly mentioning the hypotheses being tested to enhance the reader's anticipation.
I would reccomend to provide more details on how the peppermint extract was prepared and administered. Also, please clarify the rationale behind choosing the experimental dosage of 300mg/kg feed.
The results section is comprehensive and well-organized. Please, consider using subheadings to categorize findings further, especially when discussing aspects like histological analysis and molecular pathways. All these will make it easier for the reader.
I would highly reccomend organizing the discussion into subsections for clarity, such as the impact on spleen tissue, the role of peppermint extract, and the regulation of specific signaling pathways. This will enhance readability of the manuscript.
Please, enhance the manuscript by adding more reference specific studies when discussing the impact of heat stress on immune organs and the release of inflammatory cytokines. This will provide a stronger foundation within existing literature.
Please, also review the manuscript and clarify language in certain parts of the discussion for unambiguous communication. For example, when stating, "Concordantly, our study found that heat stress can significantly damage spleen tissue," provide specific findings supporting this claim.
In addition, I reccomend enhancing the content on how peppermint extract influences the NF-κB and AP-1 signalling pathways. Offering more details on specific mechanisms can enrich the discussion.
Please, do not forget to add a sentence emphasizing your study's novelty or unique contributions. When discussing the regulatory effect of peppermint extract on DAI, highlight that this is a novel finding contributing to existing knowledge.
Please, integrate references to figures within the discussion, especially when presenting novel findings or data. This will help readers visualize and understand the points being made.
In the last paragraph, you can remind readers of your initial hypotheses and objectives and discuss how your findings align or deviate from these expectations. This will provide a comprehensive interpretation of the results.
By addressing these suggestions, you can enhance the overall quality of your manuscript, making it more compelling and aligned with the journal's guidelines.
In line 214, please take into account that this sentence must be supported by a reference. In addition, Gut-Associated Lymphoid Tissue can be considered as an immune organ?
Major suggestions:
Although the approach the authors have given to the manuscript is exciting, please review the effect of heat stress on gut integrity and health. Authors will find that heat stress significantly damage the gut barrier, thus allowing the bacterial translocation from the gut lumen to the tissue. This must be considered to discuss the changes the authors have found, especially in the significantly increased cytokines and their intracellular molecular pathways. Moreover, the effect of pepermint reducing the heat stress effect on the intestinal barrier and in the microbiota must be taken into account during the discussion. Please review it, and correct it.
Comments on the Quality of English LanguageDear author, Overall, the manuscript is well-written with clear and concise language. However, I've identified a few areas where slight modifications or corrections could enhance clarity and precision:
Introduction: Please consider revising: "Chronic heat stress changes innate immune function" to "Chronic heat stress alters innate immune function."
Materials and Methods: Please consider revision of: "Ethanol was used to extract the aerial part of mentha haplocalyx in a ratio of 10:1 and peppermint extract was obtained." ("and peppermint extract was obtained" for clarity)
Results: Please consider revising: "Figure 1A was blurred, no typical splenic nodule structure was found in the tissue, vascular dysplasia occurred in a large area of the spleen, and vascular arrays are dense." ("occurred" instead of "was occurred" for proper tense)
Discussion: Please consider revising: "These results show that peppermint extract can reduce the release of inflammatory cytokines and alleviate spleen tissue injury to alleviate the inflammatory response under heat stress." to "These results indicate that peppermint extract can reduce the release of inflammatory cytokines, alleviate spleen tissue injury, and mitigate the inflammatory response under heat stress."
Conclusion: Please consider revising: "chronic heat stress activated the TLR2, TLR4, and DAI mediated signaling pathways" to "chronic heat stress activated the TLR2, TLR4, and DAI-mediated signaling pathways."
Overall, the manuscript is well-constructed, and these suggestions are minor refinements for clarity and precision.
Author Response
Response to reviewer 3
Dear reviewer:
Thank you very much for taking the time to review my manuscript. Please find the detailed responses below and the corresponding revisions highlighted in yellow of the re-submitted files.
Point-by-point response to comments and suggestions for authors
Comment 1: The well-written introduction provides a clear overview of the study's context. Consider briefly mentioning the hypotheses being tested to enhance the reader's anticipation.
Response 1: In the last sentence of the introduction, we proposed the hypothesis: “Based on the above research findings, we hypothesized that peppermint extract could regulate the innate immunity and inflammatory response through PRRs-mediated signaling pathways in spleen of chronical heat-stressed broiler chickens.”.
Comment 2: I would recommend to provide more details on how the peppermint extract was prepared and administered. Also, please clarify the rationale behind choosing the experimental dosage of 300mg/kg feed.
Response 2: In the section of Animal Management, we mentioned that the preparation of peppermint extract. The 300mg/kg was obtained from our laboratory’ s previous heat stress experiment to alleviate heat stress in broiler chickens, but it has not been published in SCI and is only included in master’s thesis (常双双.偏热环境对肉鸡盲肠菌群多样性、挥发性脂肪酸和血清脑肠肽影响及薄荷提取物对其调控作用研究[D].河北工程大学,2018.). Therefore, we did not set up groups with different concentrations of PE and directly used the most suitable addition amount obtained in the laboratory before.
Comment 3: The results section is comprehensive and well-organized. Please, consider using subheadings to categorize findings further, especially when discussing aspects like histological analysis and molecular pathways. All these will make it easier for the reader. I would highly reccomend organizing the discussion into subsections for clarity, such as the impact on spleen tissue, the role of peppermint extract, and the regulation of specific signaling pathways. This will enhance readability of the manuscript.
Response 3: The entire article is intended to serve the title of the article, so in the discussion section, we conducted a comprehensive analysis of the results. So we did not divided the discussion into several parts.
Comment 4: Please, enhance the manuscript by adding more reference specific studies when discussing the impact of heat stress on immune organs and the release of inflammatory cytokines. This will provide a stronger foundation within existing literature.
Response 4: I carefully revised the language. Regarding this sentence “Concordantly, our study found that heat stress can significantly damage spleen tissue”, in the previous sentence of this sentence, we stated that “Previous studies have found that heat stress can damage immune organ tissues [19], affect immune organ growth, and impact the overall development of broiler chickens due to decreased immune function.”.
Comment 5: Please, also review the manuscript and clarify language in certain parts of the discussion for unambiguous communication. For example, when stating, "Concordantly, our study found that heat stress can significantly damage spleen tissue," provide specific findings supporting this claim.
Response 5: I carefully revised the language. Regarding this sentence “Concordantly, our study found that heat stress can significantly damage spleen tissue”, in the previous sentence of this sentence, we stated that “Previous studies have found that heat stress can damage immune organ tissues [19], affect immune organ growth, and impact the overall development of broiler chickens due to decreased immune function.”.
Comment 6: In addition, I reccomend enhancing the content on how peppermint extract influences the NF-κB and AP-1 signalling pathways. Offering more details on specific mechanisms can enrich the discussion.
Response 6: I added the content on how peppermint extract influences the NF-κB and AP-1 signaling pathways.
Comment 7: Please, do not forget to add a sentence emphasizing your study's novelty or unique contributions. When discussing the regulatory effect of peppermint extract on DAI, highlight that this is a novel finding contributing to existing knowledge.
Response 7: In the last paragraph of the discussion, we proposed that “We found, for the first time, that dietary peppermint extract significantly decreased the mRNA expression of DAI, and decreases the mRNA expression of transcription factors (NF-κB, AP-1 and IRF3), thereby decreasing the concentration of inflammatory cytokines to inhibits activation of innate immunity and inflammatory response induced by chronical heat stress in spleens of broiler chickens.”.
Comment 8: Please, integrate references to figures within the discussion, especially when presenting novel findings or data. This will help readers visualize and understand the points being made.
Response 8: I placed my conclusion figure (Figure 5) at the end of the discussion for everyone to review.
Comment 9: In the last paragraph, you can remind readers of your initial hypotheses and objectives and discuss how your findings align or deviate from these expectations. This will provide a comprehensive interpretation of the results.
Response 9: In the last sentence of the discussion, we added “These findings also validated our hypothesis that peppermint extract could regulate the innate immunity and inflammatory response through PRRs-mediated signaling pathways in spleen of chronical heat-stressed broiler chickens.”
Comment 10: In line 214, please take into account that this sentence must be supported by a reference. In addition, Gut-Associated Lymphoid Tissue can be considered as an immune organ?
Response 10: I have revised “The spleen, as the largest peripheral immune organ, is the main site of innate immune response.” to “The spleen is an important lymphoid organ and the site to connect innate immune system cells to adaptive immune system cells [40].” and added the reference.
The question of whether gut-associated lymphoid can be considered as an immune organ is ambiguous, with some suggesting it is and others suggesting it is not.
Comment 11: Although the approach the authors have given to the manuscript is exciting, please review the effect of heat stress on gut integrity and health. Authors will find that heat stress significantly damage the gut barrier, thus allowing the bacterial translocation from the gut lumen to the tissue. This must be considered to discuss the changes the authors have found, especially in the significantly increased cytokines and their intracellular molecular pathways. Moreover, the effect of pepermint reducing the heat stress effect on the intestinal barrier and in the microbiota must be taken into account during the discussion. Please review it, and correct it.
Response 11: In the last sentence of the discussion, we proposed that “We found, for the first time, that dietary peppermint extract significantly decreased the mRNA expression of DAI, and decreases the mRNA expression of transcription factors (NF-κB, AP-1 and IRF3), thereby decreasing the concentration of inflammatory cytokines to inhibits activation of innate immunity and inflammatory response induced by chronical heat stress in spleens of broiler chickens.”.
Response to comments on the quality of English language
Comment 1: Introduction: Please consider revising: "Chronic heat stress changes innate immune function" to "Chronic heat stress alters innate immune function."
Response 1: I have revised “Chronic heat stress changes innate immune function” to “Chronic heat stress alters innate immune function”.
Comment 2: Materials and Methods: Please consider revision of: "Ethanol was used to extract the aerial part of mentha haplocalyx in a ratio of 10:1 and peppermint extract was obtained." ("and peppermint extract was obtained" for clarity)
Response 2: I have revised “Ethanol was used to extract the aerial part of mentha haplocalyx in a ratio of 10:1 and peppermint extract was got.” to “Ethanol was used to extract the aerial part of mentha haplocalyx in a ratio of 10:1 and peppermint extract was obtained.”.
Comment 3: Results: Please consider revising: "Figure 1A was blurred, no typical splenic nodule structure was found in the tissue, vascular dysplasia occurred in a large area of the spleen, and vascular arrays are dense." ("occurred" instead of "was occurred" for proper tense)
Response 3: I have revised “vascular dysplasia was occurred in a large area of the spleen” to vascular dysplasia occurred in a large area of the spleen.
Comment 4: Discussion: Please consider revising: "These results show that peppermint extract can reduce the release of inflammatory cytokines and alleviate spleen tissue injury to alleviate the inflammatory response under heat stress." to "These results indicate that peppermint extract can reduce the release of inflammatory cytokines, alleviate spleen tissue injury, and mitigate the inflammatory response under heat stress."
Response 4: I have revised "These results show that peppermint extract can reduce the release of inflammatory cytokines and alleviate spleen tissue injury to alleviate the inflammatory response under heat stress." to "These results indicated that peppermint extract reduced the release of inflammatory cytokines, alleviated spleen tissue injury and mitigated the inflammatory response under heat stress.".
Comment 5: Conclusion: Please consider revising: "chronic heat stress activated the TLR2, TLR4, and DAI mediated signaling pathways" to "chronic heat stress activated the TLR2, TLR4, and DAI-mediated signaling pathways."
Response 5: I have revised “chronic heat stress activated the TLR2, TLR4, and DAI mediated signaling pathways” to “chronic heat stress activated the TLR2, TLR4 and DAI-mediated signaling pathways”.
Round 2
Reviewer 1 Report
Comments and Suggestions for Authors
The authors have addressed most of the concerns. However, there are a few remaining issues as described below.
1. Simple summary: “Dietary peppermint extract inhibits chronic heat stress-induced activation of innate immunity and inflammatory response via DAI in spleen of broiler chickens” is overstated as there is no direct evidence.
2. Please describe how much spleen tissue the authors used for ELISA and how the authors adjusted the protein level in the Materials and Methods section.
3. Did the authors perform an equality test prior to ANOVA followed by LSD? If yes, please describe it in the statistical analysis section.
4. The authors are suggested to describe the sample size as "n = 6/group."
5. Please change "neutrophils" to "heterophils" as chickens have heterophils which are considered functionally equivalent to neutrophils.
Fingerhut, L., Dolz, G., & de Buhr, N. (2020). What Is the Evolutionary Fingerprint in Neutrophil Granulocytes? International Journal of Molecular Sciences, 21(12), 4523. https://doi.org/10.3390/ijms21124523
Stacy, N. I., Hollinger, C., Arnold, J. E., Cray, C., Pendl, H., Nelson, P. J., & Harvey, J. W. (2022). Left shift and toxic change in heterophils and neutrophils of non‐mammalian vertebrates: A comparative review, image atlas, and practical considerations. Veterinary Clinical Pathology, 51(1), 18–44. https://doi.org/10.1111/vcp.13117
6. Line 82-83 and line 311-313: These sentences are still duplicated and need to be rephrased.
7. Line 270: Please describe what “the index” means.
8. Line 276-285: These sentences are confusing as the results from previous studies and the current study are mixed up. It is highly recommended to revise them.
9. Line 319-323: This sentence is too long and overstated as there is no direct evidence. The authors need to conduct knock-in/knock-down/knock-out experiments if they want to state this.
10. Please spell out “DAI” and “TBK1”.
Comments on the Quality of English LanguageIt is highly recommended that the authors revise the entire manuscript in terms of grammatical errors and academic writing.
Author Response
Response to reviewer 1
Dear reviewer:
Thank you very much for taking the time to review my manuscript. Please find the detailed responses below and the corresponding revisions highlighted in yellow of the re-submitted files.
Point-by-point response to comments and suggestions for authors
Comment 1: Simple summary: “Dietary peppermint extract inhibits chronic heat stress-induced activation of innate immunity and inflammatory response via DAI in spleen of broiler chickens” is overstated as there is no direct evidence.
Response 1: I changed this sentence to “Dietary peppermint extract inhibits chronic heat stress-induced activation of innate immunity and inflammatory response in spleen of broiler chickens.”.
Comment 2: Please describe how much spleen tissue the authors used for ELISA and how the authors adjusted the protein level in the Materials and Methods section.
Response 2: I described the detection of ELISA as follows: “Take about 10mg of spleen tissue, weight it and put it into 1ml of physiological saline. Homogenize it on freeze grinder and centrifuge it (4℃, 3500rpm/min) for 20min. Then take the supernatant and measure cytokines according to manufacturer’s instruction. The supernatant after centrifugation was diluted to 10% with physiological saline, and the protein content was determined using the BCA protein assay kit (Beyotime, P0010). The concentration of cytokines was determined by dividing the final measured cytokine content by 10 times the protein content.”.
Comment 3: Did the authors perform an equality test prior to ANOVA followed by LSD? If yes, please describe it in the statistical analysis section.
Response 3: Yes, I have supplemented it as follows: “All data were initially tested for homogeneity of variance test and normality test.”.
Comment 4: The authors are suggested to describe the sample size as "n = 6/group."
Response 4: I changed “n=18” to “n=6/group”.
Comment 5: Please change "neutrophils" to "heterophils" as chickens have heterophils which are considered functionally equivalent to neutrophils.
Fingerhut, L., Dolz, G., & de Buhr, N. (2020). What Is the Evolutionary Fingerprint in Neutrophil Granulocytes? International Journal of Molecular Sciences, 21(12), 4523. https://doi.org/10.3390/ijms21124523
Stacy, N. I., Hollinger, C., Arnold, J. E., Cray, C., Pendl, H., Nelson, P. J., & Harvey, J. W. (2022). Left shift and toxic change in heterophils and neutrophils of non‐mammalian vertebrates: A comparative review, image atlas, and practical considerations. Veterinary Clinical Pathology, 51(1), 18–44. https://doi.org/10.1111/vcp.13117
Response 5: I changed “neutrophil” to “heterophil”.
Comment 6: Line 82-83 and line 311-313: These sentences are still duplicated and need to be rephrased.
Response 6: The sentence on Line 311-313 was changed to “The DNA-dependent activator of IFN regulatory factors (DAI), also known as DLM-1/ZBP1, functions as a DNA sensor, which can recognize DNA in the cytoplasm to activate the innate immune response mediated by NF-Κb pathway, MAPK pathway and IRF3 pathway [27, 35, 47, 48].”.
Comment 7: Line 270: Please describe what “the index” means.
Response 7: I described the meaning of the “index” as follows: “index (organ weight/body weight)”.
Comment 8: Line 276-285: These sentences are confusing as the results from previous studies and the current study are mixed up. It is highly recommended to revise them.
Response 8: I revise them as follows “Peppermint is a widely used herb, and previous studies have extensively investigated its pharmacological effects, including its anti-inflammatory immunomodulatory, antibacterial and antioxidant properties, as well as its potential mechanism [18, 50, 51]. Arab et al. (2016) investigated the effect of peppermint powder on the immune system of broiler chickens under heat stress and found that dietary supplementation of 2% peppermint powder increased IgG and IgM serum levels, and significantly increased the number of white bold cells, thereby enhancing immune function [17]. In the present study, we found that peppermint extract significantly decreased the release of inflammatory cytokines under chronical heat stress condition. Although these studies, in combination, show that peppermint has both anti-inflammatory and immune enhancement effects, relatively little work has been done on the molecular mechanism by which peppermint extract can impact innate immunity and inflammatory response. These results indicated that peppermint extract reduced the release of inflammatory cytokines, alleviated spleen tissue injury and mitigated the inflammatory response under heat stress. ”.
Comment 9: Line 319-323: This sentence is too long and overstated as there is no direct evidence. The authors need to conduct knock-in/knock-down/knock-out experiments if they want to state this.
Response 9: I changed this sentence to “We found, for the first time, that dietary peppermint extract significantly decreased the mRNA expression of DAI and transcription factors (NF-κB, AP-1 and IRF3), and decreased the concentration of inflammatory cytokines to inhibits activation of innate immunity and inflammatory response induced by chronical heat stress in spleens of broiler chickens.”.
Comment 10: Please spell out “DAI” and “TBK1”.
Response 10: I spelt out “DAI” on line 82 (The DNA-dependent activator of IFN regulatory factors (DAI)) and TBK1 on line 85 (TANK-binding kinase 1 (TBK1)).